



# Fatigue lifetime calculation of wind turbine blade bearings considering blade-dependent load distribution

**Oliver Menck, Matthias Stammler, and Florian Schleich**

Fraunhofer IWES, 21029 Hamburg, Germany

**Correspondence:** Oliver Menck (oliver.menck@iwes.fraunhofer.de)

**Abstract.** Rotating bearings are some of the most commonly employed machine elements. As such, they are well-understood and thoroughly researched pieces of technology. Fatigue lifetime calculation is internationally standardized through ISO 281, which is based on the assumption that loads act on a bearing under constant rotation. Blade bearings of wind turbines do not conform to this assumption since their movement typically consists of small, repetitive oscillations. Moreover, their load distribution differs considerably over the bearing circumference, a load case for which ISO 281 refers to ISO 16281 and which requires detailed simulations of the bearing to be sufficiently precise. Aside from ISO 16281, the NREL DG03, a guideline for pitch and yaw bearing lifetime, lists two methods for incorporating bearing loads into the fatigue life calculation. This paper compares all three methods. Two of the methods can not be used directly for the double-row four-point bearing used in this paper and are thus slightly adjusted. Load distributions in the bearing are simulated and curve-fit by means of a novel approach using regression analysis. The method from NREL DG03, which requires the least computational effort, is shown to result in a much higher lifetime than the other two, which are based on internal load distributions of the bearing. The two latter methods are shown to produce very similar results. An adjustment is proposed for increasing the accuracy of that lifetime calculation method which requires the least computational effort in order to resemble the other two more closely.

## 1 Introduction

Blade bearings are a critical component of any modern wind turbine. Enabling the turbine to pitch can reduce the loads on a multitude of its components significantly. This allows components to have a lighter design and a higher return on investment of the entire machine.

Apart from continuous pitch control (CPC), which turns all blades simultaneously by the same angle, individual pitch control (IPC) has been the subject of comprehensive research (e.g., Bossanyi, 2003; Selvam et al., 2009; Shan et al., 2013). IPC turns blades individually in order to reduce asymmetrical rotor loads, which contribute significantly to fatigue loading of a wind turbine's rotating components (Bossanyi, 2005). However, blade bearings do not necessarily benefit from this load reduction since it is achieved by increased movement of the bearing. Movements with IPC are typically small, repetitive oscillations. This movement pattern differs greatly from that of bearings in most other industrial applications, where bearings usually rotate continuously or, in some cases, turn very seldom at all. Lifetime research has hence mostly been focused on the former, which make up the vast majority of bearings sold and used. Blade bearings thus represent somewhat uncharted territory, which does not sit well with the fact that their replacement is a very costly procedure, which is thus to be avoided.

Bearings exhibit a vast number of possible failure mechanisms, including fatigue, fretting corrosion, brinelling, false brinelling, and more (Stammler et al., 2019). Of these, rolling contact fatigue of the bearing raceways used to be particularly common for rotating bearings. It has therefore been and continues to be the subject of much research (Sadeghi et al., 2009). A breakthrough was achieved by Lundberg and Palmgren (1947), who published a general calculation method for the calculation of rolling contact fatigue. Later,

and heavily based on the works of Lundberg and Palmgren, ISO 281 (ISO, 2010b) was published as an international standard for the calculation of raceway fatigue.

ISO 281 is intended for bearings under continuous rotation subjected to a constant axial and radial load or a combination thereof. To account for more complicated load situations, ISO 16281 (ISO, 2010a) was added later, allowing the calculation of fatigue lifetime for any arbitrary load situation.

Even with these standards in place, a great deal of uncertainty remained with regards to the calculation of pitch bearing lifetime. While ISO 16281 allows for consideration of the complicated load situation caused by a tilting moment, oscillatory movement patterns have yet to be considered in any of the standards. Moreover, large slewing bearings behave somewhat differently from the smaller ones on which the standard is primarily based. In 2009, the NREL CE1 (Harris et al., 2009) thus published the DG03, a guideline for the calculation of yaw and pitch rolling-bearing life. It collates the state of the art for the lifetime calculation of pitch and yaw bearings and thereby allows for the consideration of the aforementioned factors. However, none of the approaches therein has been verified for large-scale slewing bearings.

Moreover, failure modes of blade bearings are manifold and not just limited to fatigue (see Stammler et al., 2019). Wear is a typical damage mode that commonly occurs due to the small, repetitive oscillations of blade bearings, but no sufficiently reliable calculation methods exist for wear prediction in blade bearings. Hence, life calculation is typically limited to fatigue life. Assessing which failure mode is most common is difficult since, to the authors' knowledge, there are no large data sets of blade bearing failures publicly available.

The present uncertainty is reflected by the certification demands of manufacturers. In its 2003 Guideline for the Certification of Wind Turbines (Germanischer Lloyd, 2004), GL CE2 required a rating life calculation for blade and yaw bearings "if applicable". In its 2010 Guideline (Germanischer Lloyd, 2010), this requirement was removed when the guideline stated that lifetime calculation for pitch bearings was not required for any turbine. Subsequently, in the 2016 Guideline (Germanischer Lloyd, 2016), the requirement was once again changed to require a lifetime calculation according to NREL DG03 under all circumstances.

The present paper examines different lifetime calculation methods from the abovementioned standards and guidelines and compares them to each other in order to highlight differences in the methods and their results. First, the simulations underlying the present paper and the calculation methods used herein are explained in detail. Then, results of the methods presented are compared and discussed.

**Table 1.** Main turbine properties of IWT-7.5 (Popko et al., 2018).

| Property | Value |
|----------|-------|
| Rated electrical power | 7542 kW |
| Nominal rotor diameter | 163.44 m |
| Blade length | 79.92 m |
| Cut-in wind speed | 3 m/s |
| Rated wind speed | 11.7 m/s |
| Cut-out wind speed | 25 m/s |
| Minimum rotational speed | 5 rpm |
| Rated rotational speed | 10 rpm |
| Rated tip speed ratio | 7.31 |

## 2  Simulation

The calculation methods presented herein are intended for a double-row four-point contact ball bearing in a nearshore wind turbine. Turbine loads are simulated according to IEC CE3 61400-1 (IEC, 2019). The load distribution in one of the pitch bearings is simulated by means of a finite-element (FE) simulation that includes the connected blade and hub of the turbine.

### 2.1  Turbine model

Simulations of the time series were carried out using the IWES Wind Turbine IWT-7.5, a wind turbine model designed by Fraunhofer IWES and described by Popko et al. (2018). It is a nearshore turbine with 7.5 MW rated power output, designed for wind class IEC A1 (IEC, 2019). Additional properties are displayed in Table 1.

The model assumes that the turbine operates with a controller designed by the German wind turbine manufacturer Enercon. Enercon used the aeroelastic model of the IWT-7.5, equipped with their own IPC controller, to run load simulations. This controller is a wideband IPC, designed to minimize loads as much as possible without limiting the movements of the pitch bearing. The Enercon IPC activates at wind speeds slightly below rated speed. This speed region contributes a large share to the overall fatigue loads of the turbine. The control values are the loads in a nonrotating hub coordinate system, and the main objective is to minimize loads on the steel structures of the turbine (hub, machine frame, tower). Stammler et al. (2019) presented results of different load simulations with another IPC controller for the same turbine. Note that, in contrast to the present work, the controller used by Stammler et al. (2019) only activated IPC above the rated speed.

### 2.2  Load calculations

Aeroelastic simulations of the wind turbine were carried out according to IEC 61400-1, DLC1.1. A total of 20 years of lifetime were simulated. The calculated loads and move-

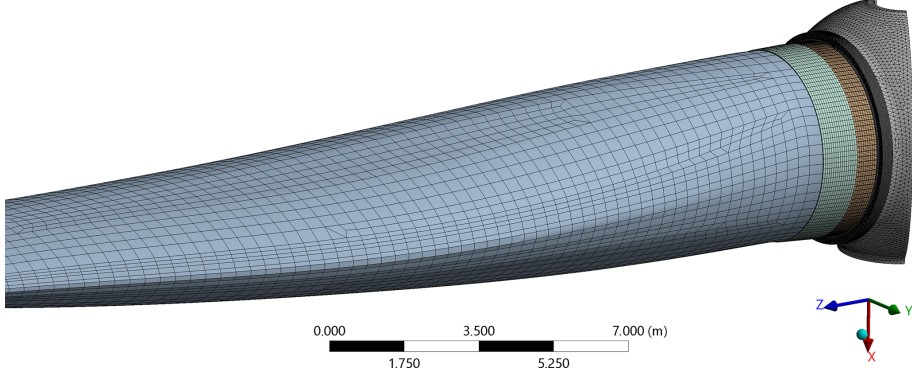

**Figure 1.** FE model of blade, bearing, and hub.

ments of one bearing are used as input for the subsequent calculations.

Available literature on the analysis of time series with oscillatory movement patterns is sparse. Fatigue lifetime calculations are most commonly done using a rain flow count (Matsuishi and Endo, 1968), which has proven to be an effective method (see Dowling, 1971). The NREL DG03 also employs a "rainbow cycle" [sic] count in one of its examples. Consequently, this paper uses a rain flow count according to ASTM CE4 E1049 (ASTM International, 2017). Note that for analyzing other types of surface-induced raceway damage, a range pair counting may prove necessary instead (Stammler et al., 2018b). While the sum of all cycles obtained using either a rain flow count or a range pair count will be identical, the length of the cycles will differ. This will have an impact on the calculated lifetime if additional factors for oscillation as used by Schwack et al. (2016) are employed.

After a rain flow cycle count of the bearing oscillations, the results were further divided into bins of the resulting tilting moment, its angle, and the absolute pitch angle of the bearing using the procedure described by Stammler et al. (2018b).

## 2.3   FE model

The generation of the entire FE model as well as all simulations was performed using Ansys R3. For all FE simulations, a one-third rotor star FE model was used. It consists of a rotor blade, one-third of a rotor hub, a pitch bearing, and a stiffener plate. Using only a one-third rotor star model greatly reduces the computational effort. Doing so makes the model behave symmetrically, meaning that it is not possible to simulate different loads acting on the three blades, a process which is assumed to have negligible effects on the loads of one bearing. Part of the model is shown in Fig. 1. The entire model consists of approximately 855 000 elements and 956 000 nodes, which results in a computational time in the range of 25 min using a PC with an Intel Xeon E5 3.7 GHz processor and 128 GB RAM.

The outer ring of the bearing connects directly to the blade flange of the hub. The bearing's inner ring connects to the blade with the stiffener plate in-between. Using stiffener plates is a common way to reduce the ovalization of the bearing, which is caused by the blade. The stiffener plate is made of steel and has a thickness of 25 mm, which is a typical thickness for a blade flange of that size. The blade model contains a fully modeled root section. The component contacts are simulated to be bonded over the connecting surfaces, meaning that no bolts and friction-based contact behavior are implemented. Not implementing bolts might lead to a little more flexibility of the bearing rings, which can result in a larger tilting of the rings towards each other. In turn the loads on both rows are distributed slightly less evenly. In-house investigations have shown that this effect only has a very small influence on the bearing's load distribution. Thus, the effect of this simplification is to be assumed negligible. The model is completely fixed at the hub's rotor flange (downwind) and partially fixed at the hub's upwind flange. In addition, cyclic constraints at the hub's one-third cutting planes are implemented. All these boundary conditions enable a realistic deformation behavior of the rotor hub to be modeled. The loads are applied to the blade's spar caps at a blade length of 40 m. Concentrating all loads acting on a blade at one point is a common method for determining bearing loads that does not completely reflect the load application on a real blade. In-house investigations beforehand have shown that doing so delivers comparable results to a realistic load application as long as the load application point is not located in the first quarter of the blade. Figure 2 exemplarily shows the characteristic deformation of the bearing rings which is caused by the surrounding structures and characterized by an imbalance of maximum deformation between the traction and compression side.

## 2.4   Pitch bearing

The bearing used for all simulations is a double-row four-point contact ball bearing as described in Stammler et al.

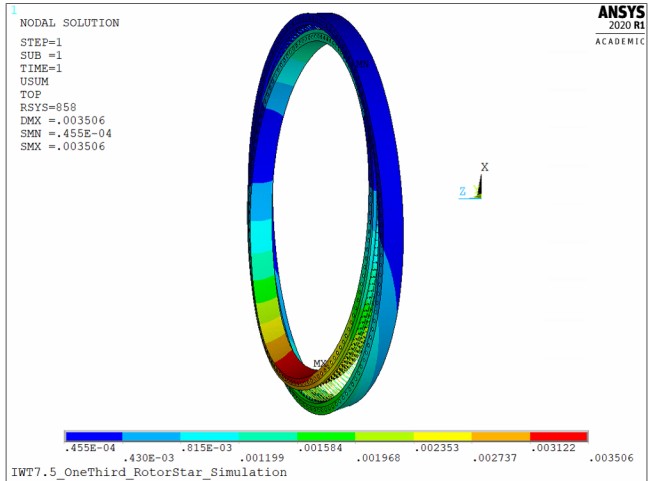

**Figure 2.** Bearing ring deformation in meters for the load case $M_y = 24\,\text{MNm}$, $F_x = 600\,\text{kN}$, $F_z = 1.2\,\text{MN}$.

**Table 2.** Main properties of blade bearing investigated (cf. Stammler et al., 2019).

| Property | Symbol | Value |
|---|---|---|
| Outer diameter | – | 5000 mm |
| Pitch diameter | $d_m$ | 4690 mm |
| Inner diameter | – | 4380 mm |
| Balls per row | $Z$ | 147 |
| Number of rows | $i$ | 2 |
| Ball diameter | $D$ | 80 mm |
| Initial contact angle | $\alpha$ | 45° |
| Total weight | – | 9232 kg |
| Load rating | $C_a$ | 3.67 MN |

(2019). Table 2 lists the main properties of the bearing. It has the typical design and dimensions of a pitch bearing for a wind turbine of that size. The pitch diameter $d_m$ refers to the distance between two opposite rolling-element centers.

The balls of the bearing are not fully modeled but represented by nonlinear springs, which is a common approach according to Daidié et al. (2008). The nonlinear force deflection curve to represent the Hertzian contact between the ball and raceway is calculated according to Houpert (2000). Figure 3 shows a cross-sectional view of the bearing model. The bearing model allows the load distribution between all balls on each of the four raceways to be analyzed.

## 2.5 Contact forces

During the aeroelastic simulations performed for the lifetime simulations of the turbine, a wide variety of different loads act on the pitch bearings. Of these, the three most influential factors are the resulting tilting moment $M$, that is, the resulting moment from edge- and flap-wise moments; the angle of said resulting moment, hereinafter called the load angle, $\beta$;

and finally the pitch angle $\theta$ of the blade. These three factors are defined as depicted in Fig. 4. They have the strongest influence on the load distribution in the bearing (Stammler et al., 2018a). While the effect of $M$ as the load onto the bearing is obvious, the influences of $\beta$ and $\theta$ are less apparent. Large slewing bearings such as those used for blade pitching tend to be very elastic. This causes the hub and blade as well as their nonrotationally symmetric stiffness behaviors to change the load distribution in the bearing significantly, depending on their respective orientation. Different load distributions in the bearing then, in turn, create a different equivalent load for the lifetime calculation. The axial force $F_z$ does not have an appreciable effect on the load distribution within the bearing as the internal axial reaction forces resulting from the tilting moment tend to be much higher. Nonetheless, a representative value of $F_z$ has been considered for the simulations, depending on the currently acting moment $M$. Radial forces $F_x$ and $F_y$ are applied at 40 m blade length as described in Sect. 2.3 and thus determined such that they cause the desired tilting moment $M$ at the blade root.

Contact forces are thus simulated at a discrete number of points using 358 different combinations of $M$, $\beta$, and $\theta$. Simulations have been run for a grid of data points, shown in Fig. 5. The grid in this case was chosen such that all operating points during the aeroelastic simulations lie within it, hence allowing a regression analysis at all times. Note that, in general, choosing a larger choice of operating points will result in more robust regression analysis results.

## 2.6 Lifetime calculation methods

All the standards and guidelines mentioned in Sect. 1 calculate rolling contact fatigue lifetime as

$$L_{10} = \left(\frac{C_a}{P_a}\right)^3, \tag{1}$$

where $L_{10}$ denotes the time until 10 % of bearings will show first signs of fatigue on their raceways (Harris, 2001). $C_a$ is the (axial) load rating of the bearing, which for all methods shown is calculated as

$$C_a = 3.647\, b_m\, f_c (i \cos \alpha)^{0.7} Z^{2/3} D^{1.4} \tan \alpha, \tag{2}$$

according to the ISO 281. $C_a$ is based on geometrical and manufacturing properties such as the row count, $i$; the number of balls, $Z$; the ball diameter, $D$; and the (initial) contact angle, $\alpha$. The factor $b_m$ equals 1.3, whereas the factor $f_c$ depends on the material and geometry of the bearing and has been chosen according to DIN SPEC 1281-1 (DIN, 2010) for all calculations in this paper. As all these parameters are known at the manufacturing stage, the load rating is generally provided by the bearing manufacturer. $P_a$ of Eq. (1) refers to the dynamic equivalent load, which is a measure for the loads acting on the bearing during operation. Three different calculation methods for determining the dynamic equivalent load

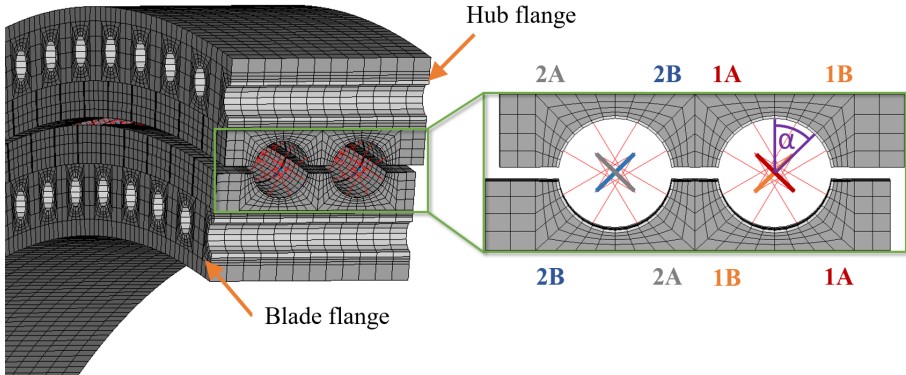

**Figure 3.** Bearing cross section and raceway definitions.

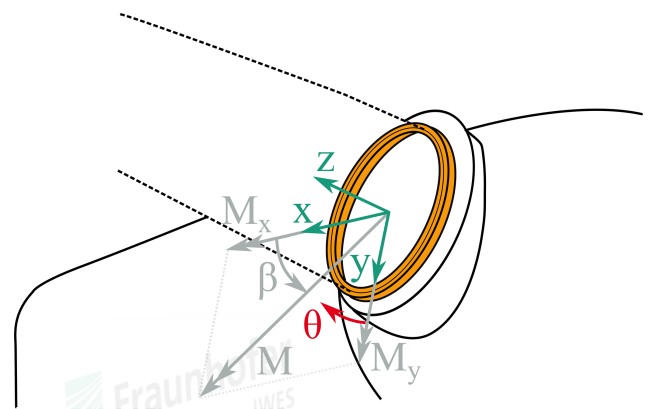

**Figure 4.** Coordinate system and angle definitions.

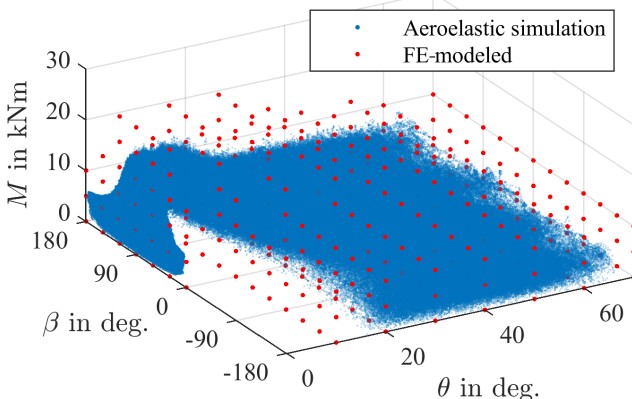

**Figure 5.** All combinations of pitch angle $\theta$, moment $M$, and load angle $\beta$ occurring during aeroelastic simulations and the combinations chosen for FE simulations.

are presented in the following, two of which, here referred to as NREL 1 and NREL 2, are listed in NREL DG03.

### 2.6.1 Method NREL 1

Firstly, an equation based on the applied tilting moment is given as

$$P_a = 0.75 F_r + F_a + \frac{2M}{d_m}, \tag{3}$$

where $F_r$ and $F_a$ refer to the applied radial and axial forces, respectively. $M$ stands for the applied tilting moment and $d_m$ for the diameter of the bearing. Equation (3) does not require any knowledge of the bearing other than its diameter and is thus often used for simple, rough calculations.

### 2.6.2 Method NREL 2

As a more sophisticated approach, the DG03 also lists

$$P_a = \left( \frac{1}{Z_{NREL}} \sum_{j=1}^{Z} Q_j^3 \right)^{1/3} \cdot Z_{NREL} \sin\alpha, \tag{4}$$

which is based on the individual rolling-element loads, $Q_j$, taken from a simulation of the entire bearing. Strangely enough, $Q_j$ is not further defined despite the fact that pitch bearings tend to be four-point bearings, meaning that up to four forces can act onto a ball at the same time. Equation (4) can thus not be used as it is shown because $Q_j$ could refer to any combination of the four contact forces. This paper therefore uses a slightly modified approach, given by

$$P_a = \left( \frac{1}{Z_{NREL}} \sum_{j=1}^{Z_{NREL}} (Q_{jA} \sin\alpha + Q_{jB} \sin\alpha)^3 \right)^{1/3} \cdot Z_{NREL}$$

$$= \left( \frac{1}{Z_{NREL}} \sum_{j=1}^{Z_{NREL}} (Q_{jA} + Q_{jB})^3 \right)^{1/3} \cdot Z_{NREL} \sin\alpha, \tag{5}$$

with the loads $Q_{jA}$ and $Q_{jB}$ as defined in Fig. 3, where effectively only the axial component of the rolling-element loads

$(Q_{jA} \sin\alpha + Q_{jB} \sin\alpha)$ is considered in the calculation. Variable $Z_{NREL}$ is interchangeably referred to as "the total number of balls in the bearing" and the "number of rolling elements in a row" by NREL DG03, even though pitch bearings are commonly double-row bearings. This paper assumes the former, meaning that $Z_{NREL} = Z \cdot i$ since otherwise no method to calculate the entire equivalent load for a double-row bearing would be defined.

Note that the variable $\alpha$ is referred to as the "nominal contact angle" in ISO 281 but simply as the "contact angle" without further specification in NREL DG03. For reasons of consistency, all calculations in this paper use the initial contact angle shown in Table 2. Contact angles of a highly loaded pitch bearing will generally increase significantly over the circumference, which would increase the equivalent load as per Eq. (5). However, consideration of a changing contact angle in the calculation of $C_a$ of Eq. (2) (and in the original calculation of $P_a$ according to Eq. 4) is not possible without changes to the overall calculation procedure and has thus not been done in the present work.

### 2.6.3   ISO 16281

As a third approach, the individual lifetime of each raceway is calculated according to ISO 16281. Like NREL 2, this approach is based on individual rolling-element loads. Equivalent loads $Q_{ei}$ and $Q_{ee}$ for a representative contact of the inner and outer ring, respectively, are calculated as

$$Q_{ei} = \left( \frac{1}{Z} \sum_{j=1}^{Z} Q_j^3 \right)^{1/3}, \text{and} \quad Q_{ee} = \left( \frac{1}{Z} \sum_{j=1}^{Z} Q_j^{10/3} \right)^{3/10}. \tag{6}$$

Note that $Q_j$ is now clearly defined as the normal force between one ball–raceway contact. All equations presented in ISO 16281 are intended for single-row bearings, while pitch bearings have two rows with four contacts on each row. Therefore the load ratings $Q_{ci}$ and $Q_{ce}$ of the inner and outer ring are calculated as

$$Q_{ci,e} = \frac{C_a(i=1)}{Z \sin\alpha} \left[ 1 + \left\{ \left[ \frac{1-\gamma}{1+\gamma} \right]^{1.72} \left[ \frac{r_i}{r_e} \left( \frac{2r_e - D_w}{2r_i - D_w} \right) \right]^{0.41} \right\}^{\pm 10/3} \right]^{3/10}. \tag{7}$$

Please note the remarks at the end of the manuscript.

$$L_{10r} = \left[ \left( \frac{Q_{ci}}{Q_{ei}} \right)^{-10/3} + \left( \frac{Q_{ce}}{Q_{ee}} \right)^{-10/3} \right]^{-9/10}. \tag{8}$$

The raceway lifetime $L_{10r}$ has thus been calculated strictly according to ISO 16281, aside from the choice of $C_a(i=1)$,

which is not explicitly demanded in ISO 16281 but can be reasoned as stated above. Finally, an adjustment of ISO 16281 is undertaken to account for the fact that the bearing has four contact pairs rather than one by calculating the total lifetime of the bearing as

$$L_{10} = \left( \sum_{p=1}^{4} L_{10r,p}^{-e} \right)^{-1/e}, \tag{9}$$

where $e = 10/9$ denotes the Weibull modulus for ball bearings (cf. DIN SPEC 1281-1; DIN, 2010). Comparing this approach to Eq. (4), it can be seen that the NREL DG03 approach is essentially an abbreviated version of the ISO 16281 using some simplifying assumptions. For informative purposes, the lifetime calculated as per Eq. (9) is then turned into an equivalent load for the entire bearing according to Eq. (1) by

$$P_a = \frac{C_a}{L_{10}^{1/3}}, \tag{10}$$

where $C_a = C_a(i=2)$ now, as usual, denotes the load rating of the entire bearing. This allows the three methods to be compared on the basis of the equivalent loads they provide.

### 2.7   Additional factors for lifetime calculations

The basic rating life $L_{10}$ as calculated according to the equations in Sect. 2.6 does not consider a variety of other factors, most notably here the movement patterns of the bearing and its lubrication conditions.

### 2.7.1   Oscillation

A number of approaches exist for factoring in the movement patterns of bearings (see Schwack et al., 2016, for a comparison). Oscillation affects the fatigue lifetime of a bearing in a number of ways, and particularly the inner (oscillating) ring will be differently loaded than on a continuously rotating bearing. Moreover, other damage mechanisms such as wear are significantly accelerated by small oscillations (see Stammler et al., 2019). With the oscillation angle $\theta_{osc}$ of a rain flow cycle bin measured in degrees, this paper assumes

$$n = n_{osc} \cdot \frac{\theta_{osc}}{180} \tag{11}$$

for all calculation methods, meaning that the various effects of oscillation on fatigue lifetime are not considered. This approach is mathematically equivalent to that presented by Harris (2001) (cf. Schwack et al., 2016).

### 2.7.2   Lubrication

Low rotational speeds worsen lubrication conditions, which, in turn, reduces the fatigue lifetime of a bearing. ISO 281

proposes an approach based on the multiplication of the basic rating life $L_{10}$ with a factor $a_{ISO}$ to receive the modified rating life

$$L_{10m} = a_{ISO} \cdot L_{10}, \tag{12}$$

which is also recommended by NREL DG03. Some differences exist in the calculation of $a_{ISO}$, but mostly the approach given by NREL is adapted from the ISO under the assumption of poor lubrication conditions. Effectively, for the simulations given, these differences have very little influence on the results. The temperature has been assumed to be 15 °C, and the corresponding viscosity of the lubricant was thus calculated according to DIN 51563 (DIN, 2011) based on two points at 40 and 100 °C. The value did not exceed $a_{ISO} = 0.1003$ at any operating point or with any method, while its minimal possible value, defined as $a_{ISO} = 0.1$, as a calculation for lower values is not possible according to the current state of knowledge as per ISO 281. Exchanging the ISO approach for that of the NREL thus only shortened the lifetime by about 0.15 % for simulations done in this paper.

## 3   Results and discussion

First, the approach for contact force regression analysis used herein is verified. Thereafter, the results of all three lifetime calculation methods listed in Sect. 2.6 are compared. An adjustment for the simplest method of all three is given so that it more closely resembles the other two.

### 3.1   Contact force regression analysis

As depicted in Fig. 5, there are significantly more data points of the aeroelastic simulation than there are FE-simulated points. A novel approach is presented to determine the contact forces $Q_j$ for all points of the aeroelastic simulation. A regression analysis of the form

$$Q_j = \left[ c_{M,0} + \sum_{k=1}^{k_{max}} c_{M,k} M^k \right]$$
$$\left[ c_{\beta,0} + \sum_{l=1}^{l_{max}} \left( c_{\beta,s,l} \sin(l\beta) + c_{\beta,c,l} \cos(l\beta) \right) \right]$$
$$\left[ c_{\theta,0} + \sum_{m=1}^{m_{max}} \left( c_{\theta,s,m} \sin(m\theta) + c_{\theta,c,m} \cos(m\theta) \right) \right] \tag{13}$$

is used for each ball–raceway contact $j$ in the bearing. Variable $k_{max}$ refers to the degree of the polynomial used to approximate moment $M$, while $l_{max}$ and $m_{max}$ denote the weights for the Fourier series used to approximate pitch and load angles $\theta$ and $\beta$. Once Eq. (13) is expanded, it is linear with regards to CE5 the various combinations of its $c$ variables. These combinations can then be determined by means of a least-square fit using results from the FE simulations.

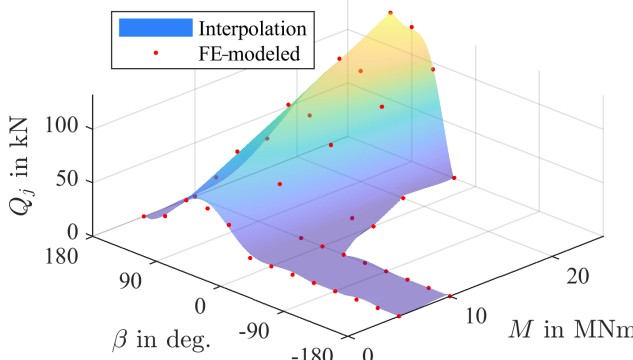

**Figure 6.** Forces $Q_j$ for a contact $j$ at 0° for a fixed pitch angle of $\theta = 10°$.

The problem being a linear least-squares problem, its solution can be determined with low computational effort using, for instance, the Moore–Penrose inverse of the problem. Even though most of its summands are not necessary for the regression analysis, the problem can thus be used as shown because the computational time remains short.

Equation (13) essentially consists of one factor for the moment, load angle, and pitch angle, respectively. While the moment is approximated with a polynomial, both angles are approximated with a Fourier series. To consider interdependencies of the three factors, they are multiplied by each other.

Figure 6 shows interpolated forces of one contact $j$ for a fixed pitch angle of 10°. Changes in the load angle $\beta$ locally take on a sinusoidal shape. For certain angles, the force disappears completely as the balls lose contact. An increase in the moment $M$ unsurprisingly leads to higher contact forces.

Likewise, the pitch angle affects the contact force as shown in Fig. 7 for another contact $j$ and at a moment of $M = 10$ MNm. As the blade pitches, stiffness properties of the inner bearing ring relative to the hub-fixed coordinate system change. This causes a different load distribution, which, for the contact shown, results in lower forces for a higher pitch angle. Contacts at other positions will behave differently.

The degree of $M$ and the weights of $\theta$ and $\beta$ should be chosen with the number of FE data points in mind. Note that with a grid as shown in Fig. 5, particular care has to be taken since in areas where the resolution is less dense (as for high moments in this example) the approximation may behave differently than in those where the resolution is higher (as for low moments in this example). Choosing $k_{max} = 3$, $l_{max} = 2$, and $m_{max} = 2$, the average error per contact in the simulated positions is then about 1100 N.

After regression analyses have been done for each $Q_j$, a contact force distribution is determined for each operating point as shown in Fig. 8 for the example of $M = 20$ MNm, $\beta = 90°$, and $\theta = 10°$. Results from the FE simulation are shown with solid lines; those from the regression analyses

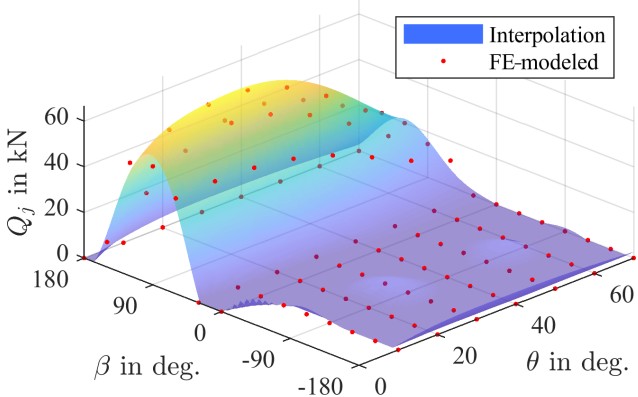

**Figure 7.** Forces $Q_j$ for a contact $j$ at $0°$ for a fixed moment of $M = 10\,\text{MNm}$.

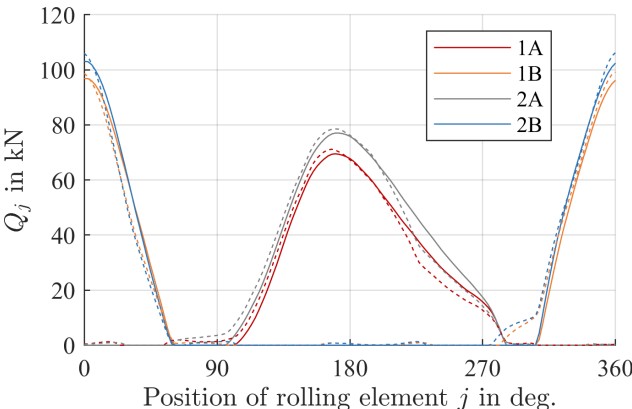

**Figure 8.** FE-simulated (solid lines) and interpolated (dashed lines) forces $Q_j$ for $M = 20\,\text{MNm}$, $\beta = 90°$, and $\theta = 10°$.

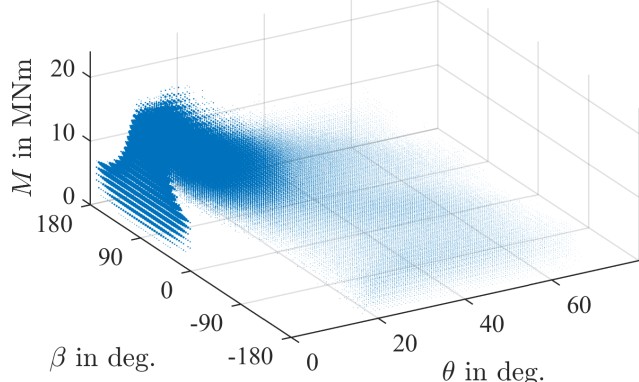

**Figure 9.** Dots for each chosen class occurring during aeroelastic simulations, with size relative to frequency of occurrence.

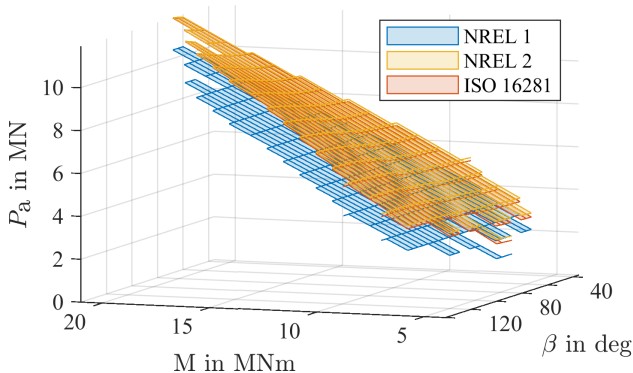

**Figure 10.** Equivalent loads $P_a$ for a fixed pitch angle of $\theta = 10°$ at operating points from the aeroelastic simulations.

performed are plotted with dashed lines. Since the calculations performed in Eqs. (4) and (6) are a type of weighted average over all contact forces, small differences in the FE simulations have negligible effects.

### 3.1.1 Lifetime calculation methods

A bin counting is carried out in order to calculate the lifetime of the bearing. For 54 different bins of the oscillation angle, the three variables moment $M$, load angle $\beta$, and pitch angle $\theta$ are put into 24, 70, and 90 bins, respectively. In total, each oscillation angle bin is thus, in turn, divided into $24 \times 70 \times 90 = 151\,200$ bins. For each of these bins, the frequency of occurrence is calculated separately. The result is shown in Fig. 9, where point sizes reflect the frequency of occurrence. Not surprisingly, most of the bearing operation takes place at low pitch angles, with $\theta = 0°$ being the most common one. Moreover, load angles between 0 and $-180°$ are also rare as these represent wind coming from behind the turbine. Consequently, it pitches out of the wind at these operating points. The maximum moment of $M = 24\,\text{MNm}$ is

only achieved at a load angle close to $\beta = 90°$ and a pitch angle close to $\theta = 10°$.

For each of the bins, the equivalent load is determined according to the three variants laid out in Sect. 2.6. Moment $M$ unsurprisingly has the strongest influence on the load. Figure 10 shows the calculated equivalent load for all existing bins with a pitch angle of $\theta = 10°$. The overall ratio between $P_a$ and $M$ can be seen to be almost linear, as assumed by variant NREL 1 (see Eq. 3). However, the calculated loads differ between the three variants. The NREL 2 and ISO 16281 variants are strikingly similar, with the NREL 2 being slightly higher in magnitude. This can be attributed to the fact that the NREL 2 method is a simplified, slightly more conservative version of the ISO 16281. Compared to the other two, variant NREL 1 is much lower: at the highest loads for $M = 24\,\text{MNm}$, this results in a difference of 12 % compared to the other two. As the equivalent load is factored into the lifetime to the power of 3 (see Eq. 1), this difference will have a considerable effect on the calculated lifetime, especially considering the fact that it occurs near a common operating point of the turbine as shown in Fig. 9.

With a constant moment of $M = 5\,\mathrm{MNm}$, the effects of the load and pitch angle can be seen in Fig. 11. As already observed above, the NREL 1 variant produces lower results than the other two. At a value of $P_\mathrm{a} = 2.48 \times 10^6\,\mathrm{N}$, it is at
5 least 13 % lower than the other two variants are at their lowest points. This difference will have a strong impact on the calculated lifetime. Moreover, it is constant for all values as it does not consider the effect of changes in $\theta$ and $\beta$ (see Eq. 3). The other two variants, however, are based on internal load
distributions of the bearing. Changes in $\theta$ and $\beta$ are thus reflected by a change in equivalent load $P_\mathrm{a}$. As can be seen, the reaction to these differences occurs with different sensitivity. For all existing bins shown, variant NREL 2 results in the highest equivalent load $P_\mathrm{a}$ since assumptions made during its
derivation from the ISO 16281 are mostly conservative in nature. The difference remains below $10^5\,\mathrm{N}$ at all times, which equals about 3 % of the maximum load for this load case. The qualitative behavior is similar as well: for both variants at $\theta = 10°$, a minimum is reached at $\beta \approx -10°$. This corre-
sponds to the situation where a moment is acting edgewise on the blade, which is why the spar caps are not carrying any load while the maximum pressure is acting on the side of the hub, which has softer stiffness behavior. With an increase in the load angle $\beta$ at the same pitch angle $\theta = 10°$, a maxi-
mum is then reached at $\beta \approx 120°$. The moment is now primarily acting flap-wise, and the spar caps are carrying most of the load, which causes the blade to exhibit stiffer behavior. Furthermore, the spar caps are pushing into the downwind side of the hub, which similarly exhibits stiffer behavior. The
overlap of the high blade stiffness due to the spar caps carrying most of the load and the stiffer backside of the hub resisting against this pressure cause the highest resulting load to occur in this position. With an increase in pitch angle $\theta$, one can see that the stiffness behavior of the blade significantly
impacts the equivalent load $P_\mathrm{a}$ as the position of the maxima and minima changes with $\theta$. At a pitch angle of $\theta = 70°$, the maximum is then closer to load angle $\beta = 20°$ since that will be the direction of a flap-wise moment.

The difference between the maximum and minimum of
40 the two methods is then approximately $5 \times 10^5\,\mathrm{N}$, which is roughly 14 % of the maximum load shown. This difference reflects the impact of $\theta$ and $\beta$. It will impact the calculated lifetime to some extent but not nearly as significantly as the resulting moment $M$.

Differences in equivalent loads between NREL 2 and ISO 16281 decrease with an increase in the moment, as shown in Fig. 12 for $M = 20\,\mathrm{MNm}$ near the highest foreseen moment of the bearing. Note that, once again, only bins that actually occurred during aeroelastic simulations are shown.
Maximum differences in equivalent loads between NREL 2 and ISO 16281 now reduce to 0.25 %, and differences caused by the load and pitch angles are at most in the range of 7 % since the turbine does not change them significantly when the resulting moments are high. This range will be the most sig-

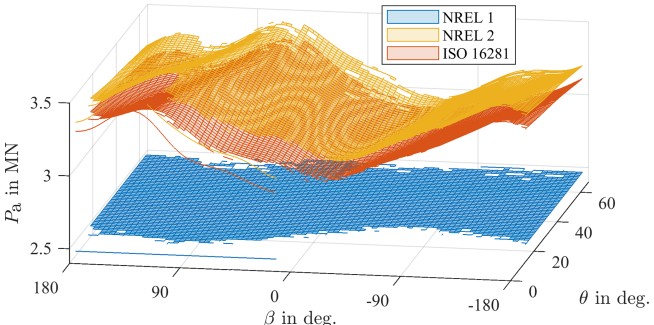

**Figure 11.** Equivalent loads $P_\mathrm{a}$ for a fixed moment of $M = 5\,\mathrm{MNm}$ at operating points from the aeroelastic simulations.

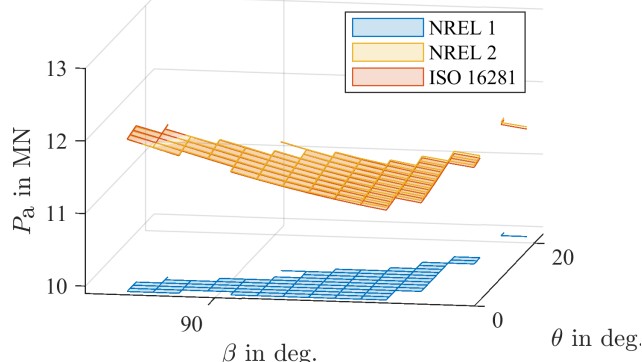

**Figure 12.** Equivalent loads $P_\mathrm{a}$ for a fixed moment of $M = 20\,\mathrm{MNm}$ at operating points from the aeroelastic simulations.

nificant for the lifetime calculation since it occurs frequently 55 and does so under high loads.

Other operating points not shown in Figs. 10 and 11 remain similar with respect to their qualitative behavior. The NREL 1 variant remains lower than the other two for every single bin examined, and equivalent loads of variant NREL 2 60 are higher than those of ISO 16281 in 99.9 % of cases.

Using these values, the overall lifetime of the bearing $L_{10m}$ is calculated, paying due consideration to the frequency of occurrence of each bin. The results are shown in Fig. 13. Additional factors have been chosen as explained in Sect. 2.7, 65 where the lubrication parameters have been used according to ISO as differences to the methods presented in NREL DG03 are negligible (see Sect. 2.7.2). As expected, the considerable differences of NREL 1 compared to the two other methods examined here have a strong impact on the calcu- 70 lated lifetime. Using variant NREL 1 for the calculation of equivalent loads hence leads to roughly 1.7 times the lifetime of the other two methods since the calculated loads are lower. Fatigue lifetime is therefore predicted to be 4273 h, or roughly 178 days. The other two methods barely differ 75 with regards to their results. Both predict a lifetime of around 107 d. This resemblance can be attributed to the similarity in

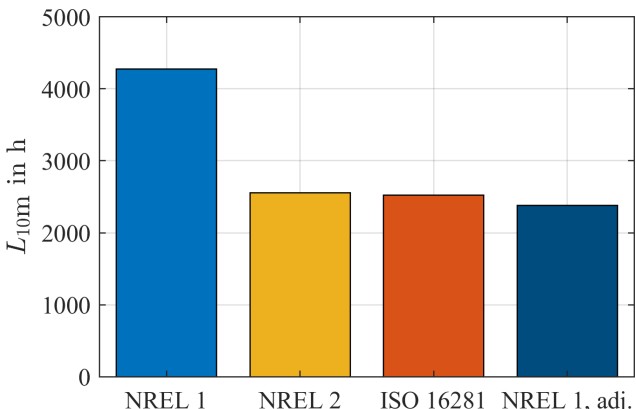

**Figure 13.** Fatigue lifetimes of all methods investigated.

their equivalent loads at common operating points of the turbine, as seen in the figures above.

These calculated lifetimes are significantly below the expected lifetime of a wind turbine of 20 years. This discrepancy is similar to other publications on the issue (cf. Schwack et al., 2016), and even calculated examples in the NREL DG03 remain well below 20 years after consideration of all factors. While available data on blade bearing failures are sparse, the calculated lifetimes are so low that no data are necessary to disprove them; the mere fact that blade bearing exchanges are costly, time-intensive operations suffices to illustrate that they will typically have to last longer.

Assessing the exact reasons for the differences between calculation and reality is, however, difficult. Calculation methods are largely based on research with small bearings, whose conditions during manufacturing and operation differ from those of large slewing bearings (cf. Göncz et al., 2010) such as pitch bearings. Large slewing bearings will, in relation to their size, generally deform more than small bearings. The stiffness of the connected structures such as the blade and hub will highly affect the deformation behavior and lead to large changes in the contact angle of 20° and more in a highly loaded state (cf. Chen and Wen, 2012), which is not considered in any calculation approach used in this paper. The usage of $a_{ISO} \approx 0.1$ may also be questioned since it reduces the calculated lifetime to a tenth of its initial value. Moreover, the lifetime $L_{10m}$ denotes the statistical point in time at which first damage occurs on the raceway for 10 % of bearings. This view of a lifetime might be too conservative for pitch bearings, which have to be as slender as possible to enable a high return on investment of the entire wind turbine and thus may continue to be operated when they are already damaged. While small bearings are expected to fail very soon after the first surfaced raceway damage, large bearings may be more robust in this regard. Lastly, as mentioned in Sect. 2.7.1, various effects of the oscillatory movement of pitch bearings are not considered in the results of this paper.

**Table 3.** Fatigue lifetimes of all methods investigated.

| Method | NREL 1 | NREL 2 | ISO 16281 | NREL 1, adj. |
|---|---|---|---|---|
| $L_{10m}$ in rot. | 14 997 | 8979 | 8844 | 8355 |
| $L_{10m}$ in $h$ | 4273 | 2558 | 2520 | 2381 |

### 3.1.2 Adjustment of NREL 1

Given the fact that the qualitative behavior of NREL 1 closely resembles that of the other two methods, as can be seen in Fig. 10, and the fact that pitch angle $\theta$ and load angle $\beta$ influence the lifetime less significantly than the resulting moment $M$, Eq. (3) provides a good basis for a simplified lifetime calculation. In order for NREL 1 to result in a similar lifetime, specifically the term $\frac{2M}{d_m}$ should be adjusted as it represents the strongest influence on the resulting equivalent load $P_a$. For the simulations in this paper, an adjustment to

$$P_a = 0.75 F_r + F_a + \frac{2.5M}{d_m} \tag{14}$$

generates lifetimes as depicted on the right-hand side of Fig. 13. The result of adjusting NREL 1 is thus slightly lower than that of its two counterparts, thereby allowing for some margin of error stemming from changes in pitch and load angle if different operating points were to occur during simulations. The calculation is, however, much simpler and thus well suited for rough analyses of the raceway fatigue lifetime of blade bearings. The results of all methods are compared in Table 3. Equation (14) is thus valid for the specific turbine examined in this paper. The authors assume that for a different turbine or, more specifically, a different combination of hub, bearing, and blade, a different adjustment may be appropriate.

## 4 Conclusions

Blade bearings of wind turbines operate under unusual operating conditions compared to others in the industry. Some details of the internationally standardized calculation of fatigue lifetime as per ISO 281, such as the calculation of equivalent loads or the consideration of oscillatory behavior, can thus be obtained by a number of methods. This paper investigated three different approaches for the calculation of equivalent loads $P_a$ required for the lifetime calculation: two according to NREL DG03 and one according to ISO 16281.

For the case of a blade bearing of the reference wind turbine, load distributions in the bearing have been simulated and interpolated to allow for consideration of a variety of operating points. The results show that changes in the load and pitch angle of a rotor blade bearing lead to significant changes in the equivalent load $P_a$. However, the impact of the resulting moment was identified to be more significant than that of the load and pitch angle.

The two methods that calculate $P_a$ on the basis of simulated load distributions (NREL 2 and ISO 16281) have been shown to provide very similar results. The third method (NREL 1), which is merely based on global loads acting on the bearing and which does not require detailed simulations of the latter, has been shown to result in much higher lifetimes than the other two methods. An adjustment has been proposed for NREL 1 to match the results of its two counterparts more closely.

As already seen in other publications on the fatigue lifetime calculation of pitch bearings, these results are far lower than the expected lifetime of a turbine of 20 years. Many reasons may account for this discrepancy: calculation methods are largely based on research with small bearings, whose conditions during manufacturing and operation differ from large slewing bearings. Effects such as the changing contact angle during operation that occurs for large slewing bearings with flexible attached structures are not considered at all. Moreover, the effect of $a_{ISO}$, which reduces the calculated lifetime to a tenth of its initial value, may be put into question. On top of that, the definition of the lifetime $L_{10m}$, which denotes the statistical point in time at which first damage occurs on the raceway for 10 % of bearings, may be too conservative for blade bearings.

**Data availability.** .TS3

**Author contributions.** OM carried out all calculations unless stated otherwise. MS wrote the tools used for data analysis and provided the idea for contact force regression analysis. FS prepared and carried out all FE simulations.

**Competing interests.** The authors declare that they have no conflict of interest.

**Special issue statement.** This article is part of the special issue "Wind Energy Science Conference 2019". It is a result of the Wind Energy Science Conference 2019, Cork, Ireland, 17–20 June 2019. TS4

**Acknowledgements.** The present work was carried out within the project "HAPT – Highly Accelerated Pitch Bearing Tests", FKZ: 0325918A. The project funding provided by the German Federal Ministry for Economic Affairs and Energy is gratefully acknowledged. Lifetime simulations of the wind turbine were created using a pitch controller from Enercon.

**Financial support.** This research has been supported by the German Federal Ministry for Economic Affairs and Energy (grant no. FKZ: 0325918A). TS5

**Review statement.** This paper was edited by Carlo L. Bottasso and reviewed by Carlo Gorla and Jonathan Keller.

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

Please note the remarks at the end of the manuscript.

## Remarks from the language copy-editor

CE1      Please define.
CE2      Should this be defined, or is it commonly abbreviated?.
CE3      Please define.
CE4      Please define.
CE5      Please confirm the change.

## Remarks from the typesetter

TS1      Please provide short title.
TS2      What does "i" and "e" indices stand for?
TS3      Please provide a statement on how your underlying research data can be accessed. If the data are not publicly accessible, a detailed explanation of why this is the case is required. The best way to provide access to data is by depositing them (as well as related metadata) in reliable public data repositories, assigning digital object identifiers (DOIs), and properly citing data sets as individual contributions. Please indicate if different data sets are deposited in different repositories or if data from a third party were used. Additionally, please provide a reference list entry including creators, title, and date of last access. If no DOI is available, assets can be linked through persistent URLs to the data set itself (not to the repositories' home page). This is not seen as best practice and the persistence of the URL must be secured.
TS4      Please confirm.
TS5      Please note that the funding information has been added to this paper. Please check if it is correct. Please also double-check your acknowledgements to see whether repeated information can be removed or changed accordingly. Thanks.
TS6      Please provide page range or article number and DOI.
TS7      Please provide DOI number.
TS8      If possible, please provide more information.
TS9      If possible, please provide more information.
TS10      Please provide all author names.
TS11      Please provide journal of publisher.
TS12      Please provide volume and page range.
TS13      Please provide DOI number or page range.
TS14      Please provide volume and page range or article number and DOI.