# Peer review of "Fatigue lifetime calculation of wind turbine blade bearings considering blade-dependent load distribution"

_Wind Energy Science, 2020_

## Referee Comment (RC1) · Jonathan Keller (Referee) · 26 Mar 2020

The authors have developed a well-written and useful paper comparing 3 different methods for the calculation of the fatigue life of wind turbine blade pitch bearings. One general comment I have is that there is no discussion of the failure modes typically seen in this application; that is, do these bearings fail by subsurface fatigue? Are there any data sources or examples that might show this? If not, then this reviewer suggests that this be discussed in the Introduction. Additional specific comments on the paper are provided below.

The Abstract and Sections 2.6.2 and 2.6.3 could be written more clearly with respect to

[Figure]

the fact that some of the methods appear to have been modified from those originally published for application to the double-row, 4-point contact ball bearings used in large, modern wind turbines. Here, by "modified", I refer primarily to the calculation of a dynamic equivalent load specific to this application. In particular for ISO 16281 in section 2.6.3, it is not clear at what point the methodology departs from the standard itself – is it equation (7) or not? The Conclusions are better written in this respect.

In the Introduction, the sentence "IPC turns blades individually in order to reduce asymmetrical rotor loads which contribute significantly to fatigue loading" would be clearer if the components or systems of interest were listed. The blades, the blade bearings, the gearbox bearings?

In Section 2.3, Figure 1 is not particularly useful as shown. If it is available, a solid body model with a cutaway view showing the components of interest would be better.

In Section 2.5, the sentence "The axial force $F_z$ has a barely noticeable effect on the load distribution as the resulting axial forces from the tilting moment tend to be much higher" could be written more clearly. I believe the intent is "The axial force $F_z$ does not have an appreciable effect on the load distribution within the bearing as the internal axial reaction forces resulting from the tilting moment tend to be much higher". Additionally, the sentence "Radial forces $F_x$ and $F_y$ result from the usage of a lever arm measuring 40m during all simulations" is not clear.

The first sentence in Section 2.6 should be "All the standards and guidelines mentioned in Sect. 1 calculate rolling contact fatigue lifetime as". Also, in equation (2), the parameter $f_{cm}$ is not defined and should be. I believe the phrase "...the figure is generally provided by the bearing manufacturer" is intended to mean "...the load rating is generally provided by the bearing manufacturer". The final sentence, I believe, should read "Three different calculation methods for determining the dynamic equivalent load will be presented in the following sections,..."

Section 3.1.1 has typos where "spare" caps should be "spar" caps. Is the statement

"...equivalent loads of variant NREL 2 are less than those of ISO 16281 in 99.9% of cases" correct? Isn't it vice versa? That is, isn't NREL 2 just slightly higher than ISO 16281? Finally for section 3.1.1, do the authors have any comment on the relatively low number of operating hours or cycles for any of the methods? What does this say about the state-of-the-art relative to pitch bearing design and existing failure rates as seen in the field? This is mentioned in the Conclusions, but the Results section would benefit from lengthier description here.

In casually glancing at the Acknowledgements, reference is made to the project "HAPT – Highly Accelerated Pitch Bearing Tests". Are there any testing results from the project that might inform or be used to validate the simulations presented in this paper? Maybe that is contained in the existing References. If so, this could be highlighted more.

---

## Author Comment (AC1) · 7 Apr 2020

First and foremost, thank you for your constructive and intelligent comments on the paper. Most of the comments were clarified in the revised paper, but there is also a reply in the following.

*The authors have developed a well-written and useful paper comparing 3 different methods for the calculation of the fatigue life of wind turbine blade pitch bearings. One general comment I have is that there is no discussion of the failure modes typically seen in this application; that is, do these bearings fail by subsurface fatigue? Are there any data sources or examples that might show this? If not, then this reviewer suggests*

*that this be discussed in the Introduction. Additional specific comments on the paper are provided below.*

Damage modes of blade bearings are manifold. Aside from fatigue, a typical damage mode is wear, which can be attributed to the small, oscillatory movements that blade bearings usually undergo. Unfortunately there is very little data - in fact, to the authors' knowledge, none - publicly available to estimate as to whether fatigue is common or not. The fact that blade bearing replacements are very expensive can only help to assess that these bearings will typically have to survive the full lifetime of a turbine.

*The Abstract and Sections 2.6.2 and 2.6.3 could be written more clearly with respect to the fact that some of the methods appear to have been modified from those originally published for application to the double-row, 4-point contact ball bearings used in large, modern wind turbines. Here, by "modifed", I refer primarily to the calculation of a dynamic equivalent load specifc to this application. In particular for ISO 16281 in section 2.6.3, it is not clear at what point the methodology departs from the standard itself – is it equation (7) or not? The Conclusions are better written in this respect*

The abstract and Section 2.6.3 have been clarified. The wording of 2.6.3 was somewhat misleading - the choice of $C_a(i = 1)$ is not explicitly demanded in the ISO 16281, since it is not intended for two-row bearings. But logically, one can easily conclude that using $C_a(i = 2)$, i.e. the load rating for the entire bearing, would not make much sense. The only "real" adjustment is the calculation of the entire bearing life as per Eq. (9). Section 2.6.2 seems sufficiently clear - the original approach is equation (4), but it cannot be used as it only considers one contact (pair) per ball, not two. Thus, equation (5) is used. Moreover, there is some uncertainty as to the meaning of $Z_{NREL}$ but that is sufficiently explained also.

*In the Introduction, the sentence "IPC turns blades individually in order to reduce asymmetrical rotor loads which contribute significantly to fatigue loading" would be clearer if the components or systems of interest were listed. The blades, the blade bearings, the*

*gearbox bearings*

IPC reduces fatigue loading mainly on the rotating components in general. Since it will generally reduce the blade root bending moment, that may have some effect on fatigue of the bearing. However, the effect on rolling contact fatigue in blade bearings is difficult to ascertain, since even though some loads may be reduced, the bearing will be subjected to more movement. Blade and hub on the other hand, for example, will clearly benefit from IPC.

*In Section 2.3, Figure 1 is not particularly useful as shown. If it is available, a solid body model with a cutaway view showing the components of interest would be better.*

Section 2.3, Figure 2 was supposed to show a cutaway view of the components of interest, which is just the bearing for this paper. The rest of the rotorstar-section model is very typical, as described in 2.3, and showing it in detail does not add much to the main topic of the paper.

*In Section 2.5, the sentence "The axial force $F_z$ has a barely noticeable effect on the load distribution as the resulting axial forces from the tilting moment tend to be much higher" could be written more clearly. I believe the intent is "The axial force $F_z$ does not have an appreciable effect on the load distribution within the bearing as the internal axial reaction forces resulting from the tilting moment tend to be much higher". Additionally, the sentence "Radial forces $F_x$ and $F_y$ result from the usage of a lever arm measuring 40m during all simulations" is not clear.*

Section 2.5, the sentence covering the axial force $F_z$ has been adjusted as proposed. The tilting moment at the root is applied using a combination of $F_x$ and $F_y$ at a constant blade length position of 40 m in z-direction, an approach that is described in 2.3. The forces $F_x$ and $F_y$ are thus determined by calculating their value so that the desired tilting moment at the blade root is achieved. This point has been clarified.

*The first sentence in Section 2.6 should be "All the standards and guidelines men-*

*tioned in Sect. 1 calculate rolling contact fatigue lifetime as". Also, in equation (2), the
parameter $f_{cm}$ is not defined and should be. I believe the phrase "...the figure is gen-
erally provided by the bearing manufacturer" is intended to mean "...the load rating is
generally provided by the bearing manufacturer". The final sentence, I believe, should
read "Three different calculation methods for determining the dynamic equivalent load
will be presented in the following sections,...*

The suggestions have been incorporated and the typo has been corrected. As for $f_{cm}$,
the load rating in this paper was calculated according to DIN SPEC 1281-1 (ISO/TR
1281-1:2008). Therein, it is called $f_c$. It depends on the geometry and material of the
bearing. This point has been clarified.

*Section 3.1.1 has typos where "spare" caps should be "spar" caps. Is the statement
"...equivalent loads of variant NREL 2 are less than those of ISO 16281 in 99.9% of
cases" correct? Isn't it vice versa? That is, isn't NREL 2 just slightly higher than ISO
16281? Finally for section 3.1.1, do the authors have any comment on the relatively
low number of operating hours or cycles for any of the methods? What does this say
about the state-of-the-art relative to pitch bearing design and existing failure rates as
seen in the field? This is mentioned in the Conclusions, but the Results section would
benefit from lengthier description here.*

The typos have been corrected. Indeed, equivalent loads of NREL 2 are higher than
those of ISO 16281, not the other way around - apologies for this confusing mistake.
The discussion of the low lifetime results has been increased in length and was put into
the Results section, at the end of 3.1.1 as follows.

"These calculated lifetimes are significantly below the expected lifetime of a wind tur-
bine of 20 years. This discrepancy is similar to other publications on the issue (cf. **?**),
and even calculated examples in the NREL DG03 remain well below 20 years after
consideration of all factors. While available data on blade bearing failures is sparse,
the calculated lifetimes are so low that no data is necessary to disprove them - the

mere fact that blade bearing exchanges are costly, time-intensive operations suffices to illustrate that they will typically have to last throughout the full lifetime of a turbine.

Assessing the exact reasons for the differences between calculation and reality is, however, difficult. Calculation methods are largely based on research with small bearings, whose conditions during manufacturing and operation differ from those of large slewing bearings (cf. Göncz et al. (2010))) such as pitch bearings. Large slewing bearings will, in relation to their size, generally deform more than small bearings. The stiffness of the connected structures such as the blade and hub will highly affect the deformation behavior and lead to large changes in the contact angle of 20 degrees and more in a highly loaded state (cf. Chen and Wen (2012)), which is not considered in any calculation approach used in this paper. The usage of $a_{\mathrm{ISO}} \approx 0.1$ may also be questioned, since it reduces the calculated lifetime to a tenth of its initial value. Moreover, the lifetime $L_{10\mathrm{m}}$ denotes the statistical point in time at which first damage occurs on the raceway for 10% of bearings. This view of a lifetime might be too conservative for pitch bearings, which have to be as slender as possible to enable a high return on investment of the entire wind turbine and thus may continue to be operated when they are already damaged. While small bearings are expected to fail very soon after the first surfaced raceway damage, large bearings may be more robust in this regard. Lastly, as mentioned in Sect. 2.7.1., the positive effect of the oscillatory movement of pitch bearings was not considered in the results of this paper."

Correspondingly, Conclusions was shortened a bit.

*In casually glancing at the Acknowledgements, reference is made to the project "HAPT –Highly Accelerated Pitch Bearing Tests". Are there any testing results from the project that might inform or be used to validate the Simulation spresented in this paper? Maybe that is contained in the existing References. If so, this could be highlighted more.*

As for results of the HAPT tests: So far, there are no results published that could be used to inform the results of this paper. Results published so far have mostly been
focused on wear (see, for instance, Bearing world journal, Volume 4_2019, Print-ISSN 2513-1753). But more results may follow.

---

## Referee Comment (RC2) · Jonathan Keller (Referee) · 6 May 2020

I also do not know of any significant, statistical published data regarding pitch bearing failure modes and rates. If the authors would like, the best that I know of is contained on slide 8 of a presentation given by Romax Technology in 2017 in which it is stated that a "Weibull analysis from one recent due diligence project indicates a 12% failure rate for pitch bearings in 20 years". This presentation is available at https://app.box.com/s/tgv9r5vpwslex2f4uh5ssk1qtwev0bxq.

I thank the authors for their consideration and incorporation of my comments to their article. I recommend that the article be published.

[Figure]

Best Regards,

Jon Keller NREL

---

## Referee Comment (RC3) · Jonathan Keller (Referee) · 20 May 2020

The reference information for the Romax Technology presentation I referred to in my prior comment was incorrect. It was a presentation given in 2019 (not 2017) and is available at https://app.box.com/s/7vo2ssy085raoeqa8wnyh5lrfjne6lgs. Best Regards, Jon Keller

---

## Short Comment (SC1) · 20 May 2020

Thank you for that presentation. While a 12% failure rate in 20 years is already much lower than calculated in the paper, it should be noted that the presentation does not even mention rolling contact fatigue (RCF) as a potential damage mode, indicating that the majority of those failures are due to other issues. RCF lifetime is, hence, even higher than an L12 of 20 years for the example given.
* * *

---

## Referee Comment (RC4) · Carlo Gorla (Referee) · 19 Jul 2020

This manuscript describes and compares three different approaches to calculate the load distribution and the lifetime blade bearings undergoing repetitive oscillations, for which the methods included in the Standard ISO 281 need to be adapted in order to take into account of different assumptions. The load calculation is based on FEM simulations. The manuscript is well organized and clearly written.

The following issues should be addressed:

1. With reference to the FEM model described by Figures 1 and 2, where the mesh is

shown, it would be interesting to have some additional information in the text (number of elements, computational time, etc.).

2. In the connection of the bearing to the mating surfaces, the effect of the bolts and of the friction forces is not considered and the component is modelled as bonded to the mating surfaces: the conclusion that the influence of this simplification is assumed to be negligible should in my opinion be supported by some additional discussion or speculation. In large bearings supported by deformable structures, the connection of the bearing to the structure can be critical. For this reason, even if this aspect does not affect the main objective of the paper, whose most relevant contribution remains the comparison among the different ways in which calculated loads are introduced in the determination of the equivalent load, some discussion on the bolting of the bearings rings and of its potential impact on the loads on the rolling element would improve the quality of the paper.

3. It would be interesting to include some contour to show the results of the simulation (e.g. stresses, displacements, etc.).)

4. I suggest to avoid the use of citations in the Conclusions. If they are relevant, it would be preferable to add and discuss them in the text, without repeating in the Conclusions.

5. In some parts of the text, as for instance in paragraph 2.3, the use of linking words is suggested to connect the sentences. Right now, each sentence is really short, and it seems to read a list of bullet points. Even if In this way the message is very clear, I would suggest a more fluid style.

---

## Author Comment (AC2) · 27 Jul 2020

Thank you very much for the criticism. The paper was updated to clarify most of the mentioned points, but there will also be a reply in the following.

*This manuscript describes and compares three different approaches to calculate the load distribution and the lifetime blade bearings undergoing repetitive oscillations, for which the methods included in the Standard ISO 281 need to be adapted in order to take into account of different assumptions. The load calculation is based on FEM simulations. The manuscript is well organized and clearly written. The following issues should be addressed: 1. With reference to the FEM model described by Figures 1 and*

*2, where the mesh is shown, it would be interesting to have some additional information in the text (number of elements, computational time, etc.).*

The entire model consists of approximately 855000 elements and 956000 nodes which results in a computational time in the range of 25 minutes using a PC with an Intel Xeon E5 3.7 GHz processor and 128GB RAM.

*2. In the connection of the bearing to the mating surfaces, the effect of the bolts and of the friction forces is not considered and the component is modelled as bonded to the mating surfaces: the conclusion that the influence of this simplification is assumed to be negligible should in my opinion be supported by some additional discussion or speculation. In large bearings supported by deformable structures, the connection of the bearing to the structure can be critical. For this reason, even if this aspect does not affect the main objective of the paper, whose most relevant contribution remains the comparison among the different ways in which calculated loads are introduced in the determination of the equivalent load, some discussion on the bolting of the bearings rings and of its potential impact on the loads on the rolling element would improve the quality of the paper.*

Not implementing bolts might lead to a little more flexibility of 100 the bearing rings which can result in a larger tilting of the rings towards each other. In turn the loads on both rows are distributed slightly less evenly. In-house investigations have shown that this effect only has a very small influence on the bearing's load distribution. Thus, the effect of this simplification is to be assumed negligible.

*3. It would be interesting to include some contour to show the results of the simulation (e.g. stresses, displacements, etc.).)*

An example of a simulation result showing bearing deformation has been included in the paper.

*4. I suggest to avoid the use of citations in the Conclusions. If they are relevant,*

*it would be preferable to add and discuss them in the text, without repeating in the Conclusions.*

The disussion of the results has been extended and citations from the conclusion have been moved to the end of the "Results and discussion" section.

*5. In some parts of the text, as for instance in paragraph 2.3, the use of linking words is suggested to connect the sentences. Right now, each sentence is really short, and it seems to read a list of bullet points. Even if In this way the message is very clear, I would suggest a more fluid style.*

The text has been changed to be more readable.